# A Novel Flow Model of Strain Hardening and Softening for Use in Tensile Testing of a Cylindrical Specimen at Room Temperature

**DOI:** 10.3390/ma14174876

**Published:** 2021-08-27

**Authors:** Mohd Kaswandee Razali, Man Soo Joun, Wan Jin Chung

**Affiliations:** 1Graduate School of Mechanical and Aerospace Engineering, Gyeongsang National University, Jinju 52828, Korea; mohdkaswandee@gnu.ac.kr; 2ERI, Graduate School of Mechanical and Aerospace Engineering, Gyeongsang National University, Jinju 52828, Korea; 3Department of Mechanical System Design Engineering, Seoul National University of Science and Technology, Nowon-gu, Seoul 01811, Korea; wjchung@snut.ac.kr

**Keywords:** tensile testing, second strain hardening parameter, second strain hardening exponent, softening behavior, damage-coupled flow model

## Abstract

We develop a new flow model based on the Swift method, which is both versatile and accurate when used to describe flow stress in terms of strain hardening and damage softening. A practical issue associated with flow stress at room temperature is discussed in terms of tensile testing of a cylindrical specimen; we deal with both material identification and finite element predictions. The flow model has four major components, namely the stress before, at, and after the necking point and around fracture point. The Swift model has the drawback that not all major points of stress can be covered simultaneously. A term of strain to the third or fourth power (the “second strain hardening exponent”), multiplied and thus controlled by a second strain hardening parameter, can be neglected at small strains. Any effect of the second strain hardening exponent on the identification of the necking point is thus negligible. We use this term to enhance the flexibility and accuracy of our new flow model, which naturally couples flow stress with damage using the same hardening constant as a function of damage. The hardening constant becomes negative when damage exceeds a critical value that causes a drastic drop in flow stress.

## 1. Introduction

Modeling of material flow stress [1,2] has been a major focus of metal-forming engineering, which not only improves our understanding of plastic deformation behavior, but also ensures high-quality engineering analyses. During bulk metal forming, flow stress data at larger strains are required, in addition to data on how cumulative damage affects the flow stress and fracture risk [3,4].

Tensile test of cylindrical specimen has many strong points in terms of material characterization, especially for bulk metal. It is simple and well-established, as well as being very reliable and reproducible. It not only gives major material properties including Young’s modulus, yield strength, tensile strength, elongation, and toughness, but also gives some implicit information about strain-hardening or strain-softening behaviors during plastic deformation of the ductile material. Note that cylindrical specimen in tensile test keeps geometric symmetry even in the region of localized necking, implying that it can experience much greater plastic deformation, which can be used to identify flow stress and reveal fracture mechanism of the materials to be metal formed.

Despite many strong points of tensile test of standard cylindrical specimens, relatively few researchers [5,6,7,8,9,10,11] until the late 2000′s had employed it to reveal the room temperature flow stresses of commercial metals at high strain. To the contrary, there had been a lot of experimental and theoretical research works on characterization of the metals over broad ranges of strain, strain rate and temperature [12,13,14,15], as well as at room temperatures using the other test methods, including the tensile test of flat or sheet materials or notched circular specimens [8,16,17,18,19,20,21,22,23]. It is to be noted that material identification from tensile test of cylindrical specimen needs extreme accuracy because of its high dependence on the necking point, complicated non-linearity, and some numerical schemes [6], which have been the factors discouraging the researchers.

Bridgman [5] made experimental efforts and tried to measure the necking profile of simple tensile test specimen. It has been practically applied to analytically correct flow behavior of a material. However, the related experiment should be carefully conducted, and its results are more or less inaccurate [9]. The traditional experimental approaches were already replaced by the digital image correlation (DIC) techniques [24,25,26,27] that freed the researchers from the difficulty in measuring the geometric features such as the curvature of the necking profile. Even though the DIC techniques have many strong points of acceptable accuracy, flexibility and applicability, the analytical or numerical approaches cannot be underestimated if their accuracy is superior to the experimental methods.

Zhang and Li [7] suggested an elastoplastic finite element (FE) analysis coupled way of predicting the tensile test of cylindrical specimen, which has a function of modifying flow stress information to trace the given tensile load–elongation curve. Even though their approach is not practical and exposed to non-negligible amount of error, they belong to the pioneering researchers in this field. Cabezas and Celentano [8] simulated the tensile tests of cylindrical and sheet specimens using the flow stress obtained from the Bridgman correction method, showing that using FE analysis of the tensile test was possible for material characterization, even though they did not use FE analysis technique directly to identify the flow stress. Mirone [9] presented an FE-based systematic and efficient approach of identifying the flow stress from tensile test of cylindrical specimen and improved the flow stress, compared particularly with the Bridgman correction method. It is, however, exposed to complexity, some inaccuracy and narrower applicability owing to the flow stress function assumed.

Joun et al. [10] took a practical approach to the acquisition of room temperature flow stresses at higher strain from the post-necking strain hardening phenomena, based on tensile testing of a simple cylindrical specimen using a finite element method [6,11,28]. They did not use any optimization technique, and the scheme can thus be perfectly separated from the FE program. The input flow stress is updated by the algorithm only once after tensile test simulation. It employs a modified Hollomon model, which can describe any kind of flow stress functions even for almost a perfectly plastic material [11] because the strength coefficient is formulated by piecewise linear functions defined by the set of sampled strains and their related strength coefficient values to be calculated. The method can be combined with any kind of finite element method, which can predict necking point exactly in the engineering sense. Eom et al. [11] showed that the model could accurately calculate flow stress even at the strain of 1.5 in case of SCM435, yielding an engineering stress–strain curve with less than 0.3% error. In addition, the method can be used to reveal the material deformation behaviors even between fracture start point and fracture point [29], which is important for understanding the fracture phenomena occurring in tensile test.

In the 2010′s, many researchers have studied the post-necking strain hardening phenomena to reveal the plastic deformation behaviors of the material, opening a renaissance of experimental-numerical combined method (ENM) and application of post-necking hardening to reveal the plastic deformation and fracture after necking. The reason lies in the fact that both the numerical methods and experimental technologies including DIC have become much advanced. It also lies in the demand from the CAE engineers or researchers who are interested in higher accuracy of finite element predictions in revealing the fracture phenomena in tensile test of cylindrical specimens or in applying new high strength materials. Literature survey on post-necking strain hardening in tensile test of standard cylindrical specimens shows that this trend has still been growing. In the early 2010′s, Gonharul et al. [30] and Donato et al. [31] defined the problem clearly and suggested an engineering measure to give further support to the determination of true material properties considering severe plasticity after necking. It is noteworthy that they emphasized properly the importance of ENM especially for experimental techniques including DIC. Eom et al. [29] coupled various damage models with the identified flow stress to evaluate the damage models in terms of fracture prediction. In the mid 2010′s, Zhu et al. [32] applied DIC technique to measure true stress–strain curves from tensile test of cylindrical specimen. Majzoobi et al. [33] presented a semi-analytical approach to correct the stress–strain curve for the large strain. Wang et al. [34] accomplished the ENM minimizing the error between the experimental and predicted load-displacement curves to describe simple tensile test after necking. Jeník et at. [35] suggested two ways, that is, ENM and a neural network approach, to identify the materials from the tensile test of cylindrical specimens. Kweon et al. [36] studied ENM based on cylindrical tensile test with well-known material models and their variants to identify the material in terms of flow stress after necking, revealing that the modified Hollomon law gives better flow stresses of SA-508 Grade 3 Class 1 low alloy steel. Paul et al. [37] suggested a simplified procedure of correcting the post-necking strain hardening behaviors based on DIC experiments with emphasis on local strain measurement at the necked region. Mirone et al. [38] presented a two-step approach to calculate the flow stress information, which translates first the engineering curves into their corresponding temporary curves called true curves using a material-independent function and then the true curves into the flow curves using the final correction function. They generally improved compared to the previous research work [9] but the error in terms of tensile load between new solutions and experiments can still be found clearly from their comparison. Chen et al. [39] studied ENM based on tensile test of cylindrical specimen using a modified Voce hardening model with emphasis on an optimization technique. They could obtain the flow curve of a mild steel with high accuracy. However, their approach is hardening model-dependent and numerical scheme-dependent and thus has some limitations or computational inefficiency; for example, it cannot deal with the localized necking phenomena, and/or it may fail to obtain an acceptable solution within a limited computational time because of its high dependence on the necking phenomena, as indicated by Tu et al. [2].

Last year, Tu et al. [2] conducted a literature survey on post-necking strain hardening and concluded that the calculated post-necking strain hardening is much depending on the material models, such as the Ludwik model [40], the Hollomon model [41], the Swift model [42], and the Voce model [43], and that the post-necking strain hardening is thus still under veil. Note that they recommended the piecewise linear hardening model for generality to cover the perfectly-plastic behaviors of the materials. They also indicated that the problem is still exposed to some numerical convergence and cost-ineffectiveness. It is, however, noteworthy that Joun’s approach evades all the problems indicated. The strength coefficient was formulated as a piecewise linear function of strain after the necking point, resulting in accurate prediction of necking point as well as versatility in describing any kind of continuous flow stress function. For example, perfectly plastic-like optimized flow stress could be expressed [6] and the maximum error in terms of difference in tensile load between experiments and predictions could be less than 0.3% [10]. It should be also noted that the approach is based not on any traditional optimization technique but on a general algorithm to improve the flow curve in an iterative way, which is developed considering the characteristics of flow stress. It is thus FEA program-independent. Four iterations are sufficient to reduce the error regardless of the number of sample points at which the flow stress should be calculated. Consequently, Joun’s approach is different from any other approaches in that it is, in fact, free from any specific flow model even though it employed a modified Hollomon model, which is more flexible in characterizing the flow curve than the piecewise linear model suggested by Tu et al. Any material model that can predict the necking point exactly can be employed for material characterization. For example, the Swift model can also be used because it can predict the necking point exactly similar to the Hollomon model. It should be noted again that accurate prediction of necking point is of great importance in identifying the flow stress using tensile test of cylindrical specimens [6], especially for application to bulk metal forming simulation.

Joun’s approach has thus a unique and important strong point that it can also be used to predict the flow stress behaviors during softening just before fracture [28]. However, the results are composed of some levels of discrete flow stress information at the sample points of strain. Our experiences say that maximum strain at the last sample point ranges from 1.0 [10] to 1.5 for SCM435 [28] or SUS304 [44], and a sort of extrapolation is thus inevitable especially for automatic multi-stage cold forging where effective strain may exceed 4.0 at locally deformation-concentrated region. Very recently, Li et al. [45] employed a special multi-camera DIC system to directly determine the true stress–strain curve over a large strain range from the tensile test of sheet specimen, revealing that there exists a distinct softening start point. It is noted that one strong point of material characterization using the tensile test is the possibility of revealing the mechanism of damage accumulation and fracture in the region of localized necking.

In this study, we start from one of the flow curves for low carbon steel obtained in the previous study [10] under isotropic hardening assumption to reveal its macroscopic behaviors with an emphasis on the localized necking just before fracture in the simple tensile test, which had scarcely been studied before. We present a modified Swift model to express the flow stress, especially for automatic multi-stage cold forging process in which the material experiences high strain. Finally, we are going to present a scheme to couple the damage with the flow stress.

## 2. Materials and Methods

Figure 1 shows an engineering stress–strain curve used in a previous study [10]; the curve is typical of those of strain-hardening materials, such as annealed mild steels cold-formed by multi-stage cold forging machines. The corresponding true stress–strain curve obtained by Joun’s approach (called the reference flow stress, denoted by RFS) is shown in the upper side of Figure 1. Only the strain-hardening region up to the true strain of 1.059 (or the corresponding engineering strain of 0.277) was examined in the previous study, which focused on predicting flow stress at higher strains described by a solid curve in Figure 1. The full curve (including the softening region ranging from 1.059 to 1.276) is indicated by the solid curve with a new dotted curve in Figure 1. Notably, Joun’s approach rapidly calculates the full true stress curve via tensile testing of the cylindrical specimen.

N, S, and F in Figure 1 correspond to the necking, softening, and fracture points, respectively; P and Q are the selected points, left and right of N’, respectively, which are necessary to calculate material constants. The first and second values within the brackets are the engineering or true strain and its corresponding stress, respectively. For example, necking occurred at an engineering strain of 0.137 and a tensile strength of 356 MPa.

The Swift flow model [46] describes flow stress as a function of strain, with material constants $Y_{o}$, $\alpha_{1}\;$ and $n_{1}$, as follows:(1)$$\sigma =Y_{o}{\left(1+\alpha_{1}\varepsilon \right)}^{n_{1}}$$
where $Y_{o}$ is the yield strength of the material, and $\alpha_{1}$ and $n_{1}$ are called the first strain hardening constant and the first strain hardening exponent, respectively. The Swift flow model and the Hollomon model [47] have been used extensively to model flow stress at room temperature. The Hollomon model relates the flow stress to the strain using a strength coefficient K and strain-hardening exponent n:(2)$$\sigma =K\varepsilon^{n}$$

The most important strain to consider when measuring flow stress via tensile testing of a cylindrical specimen is that at N [10]. The following necessary conditions should be satisfied at the necking point, i.e., for the engineering stress in tensile test of cylindrical specimen to be maximum [48]:(3)$$n_{1}=\frac{1}{\alpha_{1}}+\varepsilon_{tN}\;\mathrm{or}\;\alpha_{1}=\frac{1}{n_{1}-\varepsilon_{tN}}\;\mathrm{for}\;\mathrm{the}\;\mathrm{Swift}\;\mathrm{model}$$
(4)$$n=\varepsilon_{tN}\;\mathrm{for}\;\mathrm{the}\;\mathrm{Hollomon}\;\mathrm{model}$$
where $\varepsilon_{tN}$ is the true strain at N in Figure 1. Note that, in the Swift model, necking occurs at the same engineering strain, regardless of the values of $n_{1}$ and $\alpha_{1}$, when Equation (3) is satisfied and $\varepsilon_{tN}$ is fixed. If the flow stress at the selected point before necking (called Case 1 of Swift model, or Swift model (Case1)) must be precisely calculated, $\alpha_{1}$ and $n_{1}$ are uniquely determined. For example, the flow stress modeled by the Swift model (Figure 2a) was calculated using $Y_{o}\;$= 281.8, $\alpha_{1}$ = 101.5, $n_{1}$ = 0.137, and $\varepsilon_{tN}$ = 0.127, thus satisfying the RFS of 322 MPa at the strain of 0.016 (at the selected point “P” in Figure 1). It should be noted that the sensitivity to the strain at the selected point when calculating $\alpha_{1}$ for Swift model (Case 1) is very high; for example, when the sample point increases from 0.016 to 0.071, $\alpha_{1}$ decreases by 80%, implying that Swift model (Case 1) is unstable.

In the extreme case of a Swift model with an extremely large value of $\alpha_{1}$, that is, the Hollomon model, the strain-hardening exponent n should be $\varepsilon_{tN}$, as in Equation (4) (the Considère criterion [48]), and the strength coefficient K can be determined to characterize the true stress in the tensile test at N in Figure 1. Note that the Hollomon and Swift curves are almost the same, as can be seen in Figure 2a. It was previously shown [10] that flow stress in the Hollomon model, calculated with n = 0.127 and K = 526.4 MPa, may be significantly underestimated at the larger strain after N, as shown in Figure 2a. It is because the increased dislocation density in the diffuse necking region reinforces the forest strain hardening at room temperature. But the Hollomon model and the like, with fixed constants, cannot deal with this phenomenon.

Figure 2a emphasizes that the flow stress of the Swift model used in Case 1 is also significantly underestimated at the strain after N compared to the RFS, which is very close to experimental tensile behavior predictions [6], as can be seen in Figure 2b. As a consequence, both the Hollomon model and the Swift model (Case 1), which emphasize the strain before N, may be inappropriate for characterizing the flow stress of common metallic materials in terms of the RFS. The importance of the strain after necking to bulk metal forming cannot be overemphasized, especially in the context of forging, because of its larger strain compared to sheet metal forming. Necking is of great importance for characterizing the plastic deformation of material; this motivated previous research [10] that aimed to derive a practical method for expressing the strength coefficient as a function of strain after N.

Figure 2b compares the experimental tensile test results [6] with the predicted behavior [49], based on the flow curves shown in Figure 2a. It is noteworthy that the maximum error of the predicted tensile load with the RFS was less than 0.03% of the experimental tensile load [10], between N and the start point of softening, S, and was 2.2% between S and F (Figure 1). The three flow curves in Figure 2a all meet at N. However, all of the curves differ from the experimental curves to some extent; the RFS predicts the tensile test outcomes with acceptable accuracy, despite small differences in the region before N. The other engineering stress–strain curves bend slightly more than the experiment, as expected [8,9,50]. The errors of the Swift model (Case 1) and the Hollomon model at an engineering strain of 0.26, compared with an experimental engineering stress of 318 MPa, are 11.6% and 16.0%, respectively; flow stress is underestimated after N (Figure 2a).

If the flow stress before N is neglected in the Swift model, more accurate estimation of the flow stress after N (called Case 2 of Swift model or Swift model (Case2)) can be achieved. For example, the RFS value of 556 MPa at the strain value of 0.753 at the point selected on the right side, denoted by “Q” in Figure 1 or Figure 3, together with Equation (3), gives $Y_{o}$ = 339.6, $\alpha_{1}$ = 7.39 and $n_{1}$ = 0.262; its flow curve is shown in Figure 3a, indicating that it can depict the RFS after N relatively accurately up to S. In Figure 3b, the tensile behavior predictions based on the flow stress of the Swift model (Case 2) in Figure 3a are compared with the experimental results, indicating that the true stress after N can be predicted with acceptable accuracy at the expense of the low strain before N. In addition, as $\alpha_{1}$ decreases, $n_{1}$ increases from Equation (3), or strain hardening at the larger strain can be overestimated, implying that strain hardening can be controlled by $\alpha_{1}$ with little loss of accuracy before N.

However, flow stress before and after N in Figure 2a cannot be considered simultaneously by the original Swift model when the necking point should be accurately predicted. In many cases, including drawing or extrusion process and sheet or plate metal forming process in which strain is not so great or necking phenomenon does occur, the flow stress before N is also important. To resolve this, i.e., to describe the flow stresses before and after N with acceptable accuracy, as well as to predict N accurately, additional material parameters are essential. The following flow model, which is an improved Swift model, is thus proposed:(5)$$\sigma =Y_{o}{\left(1+\alpha_{1}\varepsilon +\alpha_{2}\varepsilon^{n_{2}}\right)}^{n_{1}{}^{\prime }}$$
where $\alpha_{2}$ and $n_{2}$ are the second strain hardening parameter or constant and second strain hardening exponent, respectively. $n_{1}{}^{\prime }$ is a variant of $n_{1}$ in Equation (1), called the first strain hardening exponent in the present model.

The term $\alpha_{2}\varepsilon^{n_{2}}$ should not have a large effect on flow stress in cases with small strain, but its effect should become greater for larger strain. Note that the above requirements can be fulfilled with a larger value of $n_{2}$. When the present flow model is employed, necking occurs when the following condition holds:(6)$$n_{1}{}^{\prime }=\frac{\alpha_{1}{}^{-1}+\varepsilon_{tN}+r\varepsilon_{tN}{}^{n_{2}}}{1+n_{2}r\varepsilon_{tN}{}^{n_{2}-1}}$$
where $r=\alpha_{2}/\alpha_{1}$. The above condition was derived from the necessary condition of maximizing pull force P in the tensile test, which is formulated as follows:(7)$$P=\frac{A_{o}Y_{o}}{1+\varepsilon_{eN}}{\left(1+\alpha_{1}\varepsilon_{tN}+\alpha_{2}\varepsilon_{tN}{}^{n_{2}}\right)}^{n_{1}{}^{\prime }}$$
where $\varepsilon_{eN}$ and $\varepsilon_{tN}=\ln(1+\varepsilon_{eN})$ are the engineering and true strains at the necking point, respectively, $A_{o}$ is the initial cross-sectional area of the cylindrical tensile test specimen, and $Y_{o}$ is the yield strength.

It is noteworthy that $n_{1}{}^{\prime }$ is almost the same as $n_{1}$ in the Swift model in cases with a larger value of $n_{2}$. For example, the error between the terms is around 0.3% when $\alpha_{1}$ is the same as in the Swift model (Case 1), and when $\alpha_{2}=8.0$ and $n_{2}=3.0$. It is thus recommended that Equation (6) be replaced with Equation (3) for simplifying engineering calculations under the condition that $n_{2}$ is greater than 3.0, implying that $n_{1}{}^{\prime }$ can be approximately calculated in the same way as $n_{1}$ in the Swift model. Figure 4 shows the effect of $n_{2}$. In cases where a value of 3.0 is used for $n_{2}$ and the true strain $\varepsilon_{tN}$ = 0.127 at N in the tensile test (see Figure 1), the difference in $n_{1}$ between Equations (3) and (6) ranges from 0.3% to 4.5% for values of $\alpha_{2}$ in the range 8–100, which can be neglected for engineering applications. The error can be reduced by modifying the RFS curve or by directly solving the equations.

The parameter $n_{2}$ with $\alpha_{2}$ can thus have a meaningful effect on the true stress only when the strain is much higher than that at N in Figure 4. With $n_{2}$ fixed at 3.0 based on numerical results, the flow stress pattern can be controlled by $\alpha_{2}$ alone, as shown in Figure 5a. Figure 5b shows the tensile behavior predictions derived under various values of $\alpha_{2}$, with $n_{2}$ fixed at 3.0; the new term $\alpha_{2}\varepsilon^{n_{2}}$ allows for a more flexible flow model. It shows that under different $\alpha_{2}$ values (see Figure 5a), necking occurs at almost the same engineering strain, as discussed above, even though there may be instability [10] due to the existence of many local maximum points. The $\alpha_{2}$ value of the second strain hardening parameter or constant can be determined according to the experimental flow stress or RFS at a selected point, or to predict the forming load of a reference forming process. For example, the flow stress denoted by “R” in Figure 5a was defined by $\alpha_{2}$ = 155 and $n_{2}$ = 3, such that the calculated flow curve passes through the point marked “Q” on the RFS curve in Figure 1 or Figure 3a. Figure 6a compares the flow stress calculated by the present approach denoted by R with Case 1, Case 2, and RFS; the flow curve R passes through point Q on the RFS curve, and thus matches the RFS curve more accurately compared with Cases 1 and 2 after the necking point N. The maximum error between the experiment and the Swift model (Case 1), however, amounts to 19.8% at S. The tensile predictions with flow stresses of RFS, Swift (Case 1 and 2), and the present model in Figure 6a are compared with the experimental results in Figure 6b, which shows that the present flow stress predicts the tensile test with acceptable accuracy from point P to point S. The error of the predicted engineering stress in this range is less than 1.5%. However, the engineering stress is underestimated by RFS, especially around point P (before N). The maximum error of 4.2% occurs close to point P. It should be noted that the RFS curve was obtained via four iterations and described as a piecewise function with 30 coefficients defined at 15 sample points, i.e., non-closed-form function; thus, the drawbacks described previously still exist. The Swift model exhibits large flow stress error before (Case 2) and after (Case 1) N compared with the other flow stresses. It is concluded that the present approach can deal with only one (when $n_{2}$ is fixed at 3) or two additional constants or parameters to describe the strain hardening behaviors of materials with high accuracy and flexibility. It is noteworthy that the additional variables deem to reflect the increasing dislocation density into the strain hardening behavior of materials, i.e., forest strain hardening.

## 3. Description of Strain Softening Using the Second Strain Hardening Parameter

Material strain softening takes place during metal forming of ductile materials at room temperature due to the accumulation of damage [3,4,45,50] with plastic deformation or increased strain. About four decades ago, Gurson [51] studied the role of hydrostatic pressure in plastic deformation with emphasis on void nucleation and growth for porous ductile materials using the upper bound approach, revealing that plastic dilatation depends on porosity and that porosity can be nucleated during straining at second phases in a ductile matrix or at grain boundary misfits. Tvergaard and Needleman [3] and Gelin et al. [4] numerically solved the strain softening phenomena as the problems of void nucleation and growth reflecting the damage effects in tensile test of standard cylindrical specimen and notched cylindrical specimen, respectively. Both of the studies clearly stated that the void volume fractions or strain softenings increase rapidly to give rise to fracture. Recently, Li et al. [45] verified the distinct strain softening behavior occurring in tensile test before the fracture point using multi-camera DIC system.

In this section, the above softening phenomena are to be expressed in a mathematical form, which can be used for simulating material test and metal forming. When the strain softening was neglected, the error in the engineering stress shown in Figure 6b increased gradually after S, reaching 4% at F. Figure 1 shows the remarkable change in flow stress after S, which is believed to result from cumulative damage. The RFS flow curve in Figure 1, which reflects the effect of the softening as a function of strain, predicts the tensile behavior relatively accurately up to F, as can be seen in Figure 2b, Figure 3b and Figure 6b.

Unfortunately, the damage-uncoupled flow stress function in Figure 1 cannot be generally applied; it can be used only for the example tensile test because damage accumulation differs according to the stress state variables and damage model used [32,52,53,54,55,56,57,58,59,60,61,62]. Therefore, an accurate damage-coupled flow model, independent of the damage model and able to predict tensile behavior with an emphasis on the softening, is needed. Various methods have been used to capture the effect of damage on flow stress [63,64,65,66,67]. However, very few practical quantitative methods have been realized, because of the complexity of the equations and difficulty in acquiring material constants. There has been no successful attempt to quantify the softening phenomenon described above.

Figure 5a shows that lower strain hardening can be described by using a smaller value of $\alpha_{2}$. As an extreme case, a negative value of $\alpha_{2}$ reduces the flow stress as the strain increases, as shown in Figure 7. Various softening phenomena can be described by calculating $\alpha_{2}$ in terms of strain and/or damage, to reflect the softening under conditions of larger strain or greater damage. For example, $\alpha_{2}$ may be formulated as a function of damage, as follows:(8)$$\alpha_{2}=\alpha_{2o}\left[1-q\frac{\langle R_{CR}-p\rangle }{\left(1-p\right)}\right]$$
where $\alpha_{2o}$ is a constant for the base material (the second strain hardening parameter); *p* and *q* are input constants; $R_{CR}=D/D_{cr}$ where D and $D_{cr}$ are the cumulative and critical damage, respectively. The function 〈x〉 has a value of x if x is positive and a value of 0 if x is negative or zero.

Note that *p* and *q* can be determined to reflect post-necking phenomena or strain softening. It is interesting to note that the flow stress starts to drop from point S, i.e., before F, as shown in Figure 1, thus implying that strain hardening stops naturally at that point, which is also deemed the starting point of macroscopic coalescence of pores due to cumulative damage [68]. It is recommended that the *p*-value be determined according to the ratio of engineering strain at S to that at F, as in Figure 1. For the example provided in this study, *p* = 0.85 was calculated based on engineering strains of 0.277 and 0.327 at S and F, respectively. The *q*-value was determined as 2.0 based on trials to fit the predicted tensile behavior with the experimental results.

Using the above conditions and a normalized Cockcroft–Latham damage model [52], finite element prediction of tensile behavior up to F was made using a rigid-plastic finite element method with a quadrilateral-element based-half perfect analysis model without any imperfection [6], as shown in Figure 8 and Figure 9. Note that the elastoplastic finite element method needs a kind of imperfection such as tapered shape, localization, artificial strain, and so on. Figure 8 compares the predictions made by the present damage-coupled flow model against the experimental results; the present flow model is relatively accurate, especially in the softening region, i.e., its maximum error is less than 0.4% near F, while the maximum error of the RFS is around 2.4%. It should be emphasized that the accuracy of the tensile behavior predictions in this study are highly significant, because the results could be employed for evaluating the ductile fracture of materials during actual metal forming. Figure 9 shows the variation of effective stresses at the selected point with the stroke during the tensile test; softening starts from the central axis and moves toward the outside of the tensile specimen at F. Note that the effective stress in necking region is the same with the flow stress because plastic deformation always occurs in this region. Figure 10 shows that the trajectory of flow stresses with respect to strain differs among sample points, due to non-uniform cumulative damage caused by stress non-uniformity or due to triaxiality after necking. Because the central N, where all stresses except axial stress disappear, is exposed to the most severe damage in the normalized Cockcroft–Latham damage model, the minimum flow stress with respect to strain was observed at this point. Note that the minimum flow stress of 100 MPa was applied for predicting the tensile behavior to alleviate the numerical problem of zero-flow stress. The assumed minimum flow stress value had a negligible influence on the tensile behavior predictions.

To verify the *p* and *q* values, four different damage models, including the normalized Cockcroft–Latham [52], and models by Brozzo et al. [53], Oyane [54], and Rice-Tracy et al. [55], with *p* and *q* taking fixed values as above, were tested in terms of engineering stress. The results are shown in Figure 11, and it can be seen that they are almost the same, except for the point immediately before the fracture, where numerical instability exists. It is therefore concluded that *p* and *q* are independent of the damage model used especially in tensile test and thus can be considered as tensile test-based material properties representing the effects of damage with emphasis on localized necking. It is also interesting to note from Figure 11 that the present damage-coupled model can describe the sudden decrease in pull force at F owing to the features of negative $\alpha_{2}$, second strain hardening parameter.

## 4. Discussions and Conclusions

### 4.1. Discussions

Accuracy of finite element predictions of metal forming processes depends dominantly on flow behaviors together with tribological factors. Even though discrete or piecewise description methods of the flow behaviors are quite practical and flexible especially for numerical analyses [10,69,70], the closed-form function models including the fundamental flow models of Ludwik [40], Voce [43], Hollomon [41], and Swift [42] are still attractive from many viewpoints. For example, they cannot only give us some insight into the material behaviors but they are also numerically strong and extensible enough to deal with the strain staying out of reach of measurement.

However, before Joun et al. [10], they could meet only the flow curve at lower strain, as shown in Figure 12 (The fundamental flow curves were fitted using the necking conditions and/or yield strength), even though the flow curve at post-necking strain could be approximately calculated from the measured curvature [11]. Note that such an approximate flow curve may incur quite a great error in terms of predicted tensile test and that it can be hardly used to reveal the fracture mechanism around the fracture point, that is, from Point S to Point F in Figure 1. It should be also noted that the facture occurring in tensile test is of great importance, and a study on it should precede that of any other fracture case because of its geometric and mechanical simplicity [29,71]. Nonetheless, this fundamental problem could not be easily tackled because of the extreme sensitivity of the plastic deformation in tensile testing to the necking. This high sensitivity had prohibited the researchers from obtaining the flow curves at large strain using traditional optimization techniques [2] before the combined experimental and numerical approaches using DIC technologies were much developed.

Many researchers have modified the fundamental flow models. In particular, the Voce model was modified by a few researchers. However, they neglected the existence of its peak curvature point between the near-linearly increasing region at low strain and asymptotic line at large strain. It can be thought that most Voce family models employed the asymptotic line side if we view them from the standpoint of large strain for bulk metal forming. This fact means that they are still inefficient even though the number of their material constants increased. To the contrary, the present flow model can make it satisfactory to meet yield point, necking point with the Considère criterion, and a flow point on the post-necking region at once. Therefore, the present flow model can be first used to describe the flow curve for predicting the tensile test with high accuracy considering the strain softening just before the fracture point. Note that the discrete or piecewise description of flow curve in the strain softening region may cause some numerical uncertainties including non-convergence or numerical oscillation in fixing the unknown flow curve.

It should be, however, noted that the strain hardening may be severe at large strain (>3.0) in case of large second strain hardening parameter, as can be shown in Figure 4, because the second strain hardening exponent (3.0–4.0) should be great enough to make Equations (6) and (7) meaningful. We suggested a scheme of coupling the second strain hardening parameter with damage because the cumulative damage increases with deformation or strain, as described in Equation (8). Even though it was originally proposed to describe the softening phenomena before the fracture point in tensile test, it can be used to describe the strain softening owing to the increase in strain as well as in damage.

It should be emphasized that only the second strain hardening parameter with fixed second strain hardening exponent can control the flow curve at the post-necking strain with negligible or minimized influence on the flow behaviors before the necking point. The present flow model is thus quite practical when we should consider the uncertainty of flow curve at large strain in the procedure of process design in cold metal forming without any meaningful damage of the flow curve at low strain. It can be also effective in the cases that either the flow stresses at both pre- and post-necking strains are important at once or necking and/or fracture may occur during plastic deformation of materials.

### 4.2. Conclusions

The Swift and Hollomon flow models at room temperature, which include three and two material constants, respectively, were investigated. Usefully, both models can predict the necking point N using material constants easily derived from experimental tensile test curves. However, the constants employed did not adequately reduce the pull force difference between the model equations and the experimental/reference curves, because two parameters are required to identify N. It is thus impossible to calculate accurate flow stresses before and after N simultaneously with the only one parameter of the Swift model left out of three constants.

We exploited the key advantages of the Swift model (easy and accurate prediction of N and broad applicability) when developing a new flow model. We used a special higher power strain term, derived by multiplying a second strain hardening exponent power to strain with a second strain hardening parameter. The value of the exponent is recommended to be greater than 3 to ensure that it has a negligible effect under conditions of small strain with respect to the calculation of N. This is very important when determining the flow stress of metallic materials at room temperature. The new term enhances the flexibility and accuracy of flow models at higher strains; relatively accurate predictions of tensile behavior are possible for regions before and after N, and at N itself. This additional term can play a role of reflecting the increasing dislocation density onto the strain hardening, particularly in the diffuse necking region.

A negative second strain hardening parameter drastically reduces flow stress when strain exceeds a critical value; this naturally couples flow stress to damage, where the second strain hardening parameter is a function of cumulative and critical damages. A method of determining the function was also proposed. Our method can simply determine the extent of damage because it uses only two parameters, which are also independent of the damage model used in tensile test; these parameters can be viewed as tensile test-based material properties describing the effects of damage (softening) on flow stress with emphasis on the localized necking.

In conclusion, the presented novel flow model, based on the well-known Swift model and supported by the generalized Hollomon model [10] to describe the flow stress with high accuracy and generality, can be used to easily obtain accurate material constants based on the tensile test data and express the related flow stress in a compact form considering strain softening or damage effect as well as the varying strain hardening owing to increasing density of dislocations. The model will be of practical utility for metal forming and improves our understanding of entire macroscopic phenomena occurring during tensile testing, including even strain softening due to damage and fracture as well as strain hardening with increasing dislocation density effect considered.

## Figures and Tables

**Figure 1 materials-14-04876-f001:**
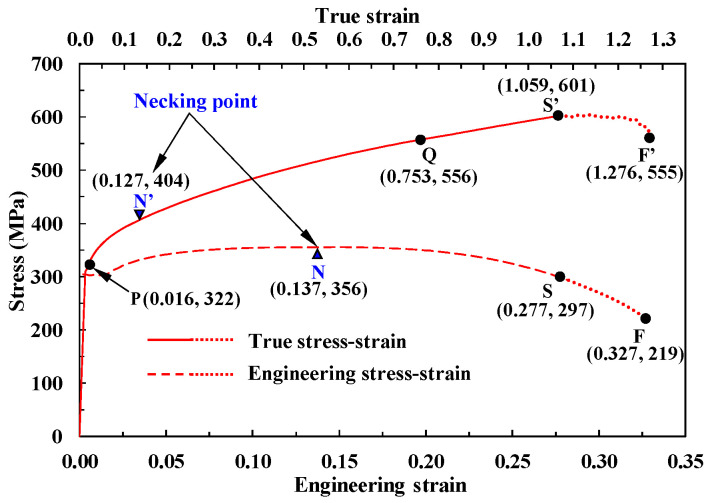
Experimental engineering stress–strain curve of SWCH10A and its associated true stress–strain curve (RFS) [10]. The dot curves were considered as the ductile fracture-effective region.

**Figure 2 materials-14-04876-f002:**
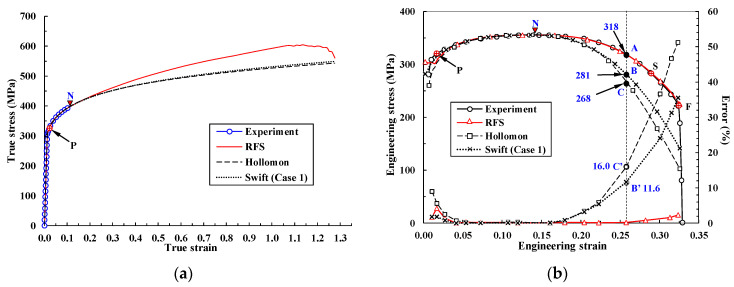
Flow curves of RFS, Swift model (Case 1), and Hollomon model and comparison of their predictions of the tensile test with the experimental tensile test. (**a**) True stress–strain curves; (**b**) engineering stress–strain curves, plotted with error percentage.

**Figure 3 materials-14-04876-f003:**
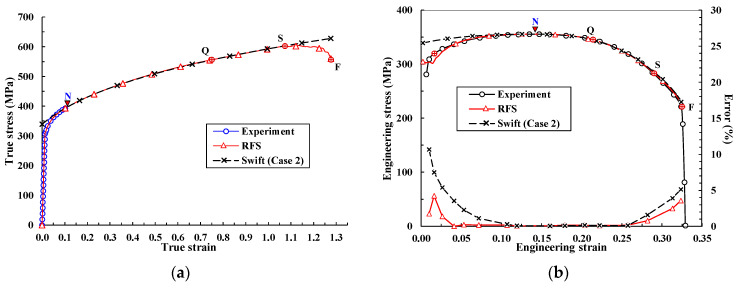
Flow curves of RFS and the Swift model (Case 2) and comparison of their predictions of the tensile test with the experimental tensile test. (**a**) True stress–strain curves; (**b**) engineering stress–strain curves, plotted with error percentage.

**Figure 4 materials-14-04876-f004:**
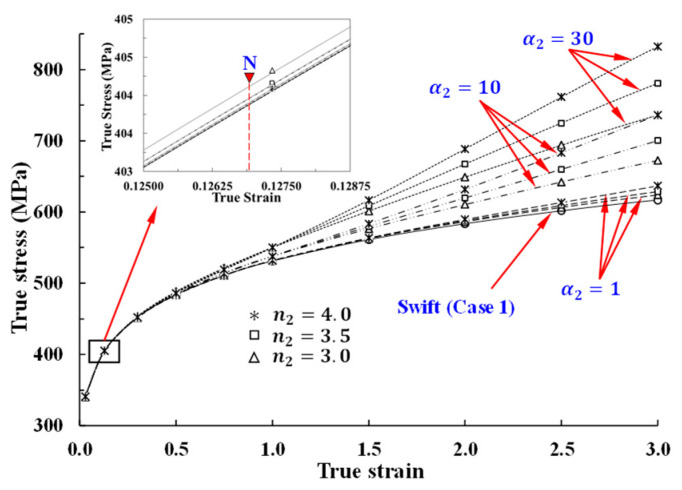
Effect of $n_{2}$ on the flow curve.

**Figure 5 materials-14-04876-f005:**
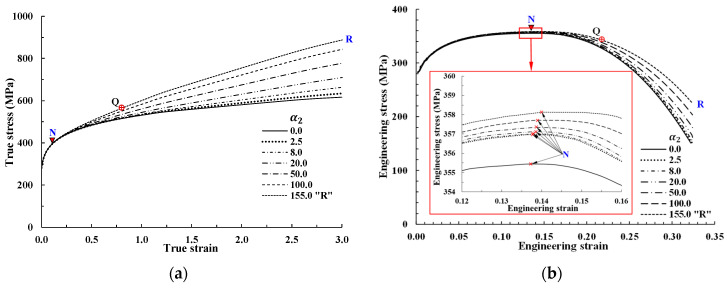
Flow curves and their corresponding tensile behaviors at various $\alpha_{2}$ values with $n_{2}$ fixed at 3.0: (**a**) Flow curves controlled by $\alpha_{2}$ when $n_{2}$ is fixed at 3.0; (**b**) predicted tensile behavior according to the flow curves in Figure 5a.

**Figure 6 materials-14-04876-f006:**
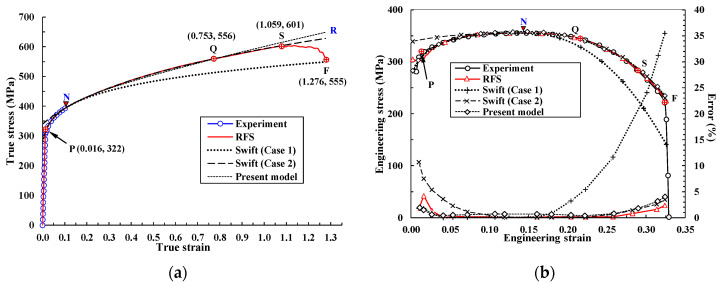
Flow curves of RFS, Swift models (Cases 1 and 2), and the present model and comparison of their predictions with experimental tensile test: (**a**) True stress–strain curves; (**b**) engineering stress–strain curves.

**Figure 7 materials-14-04876-f007:**
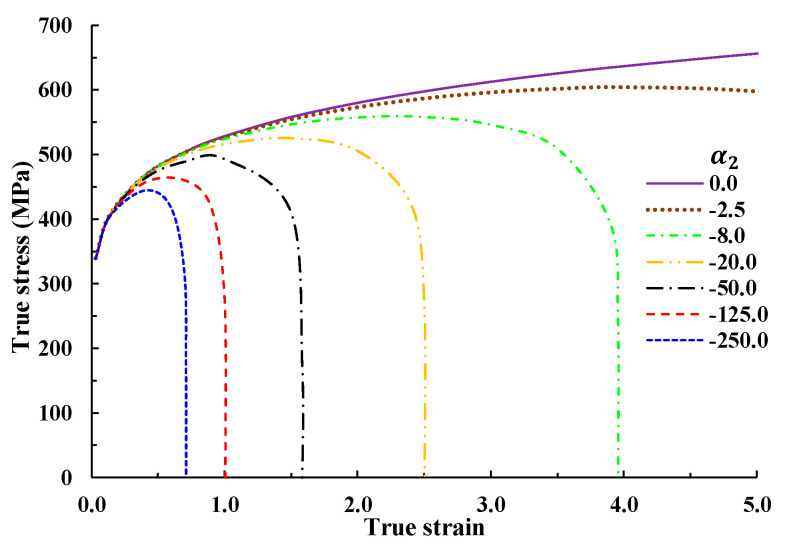
Flow curves derived using negative $\alpha_{2}$ values.

**Figure 8 materials-14-04876-f008:**
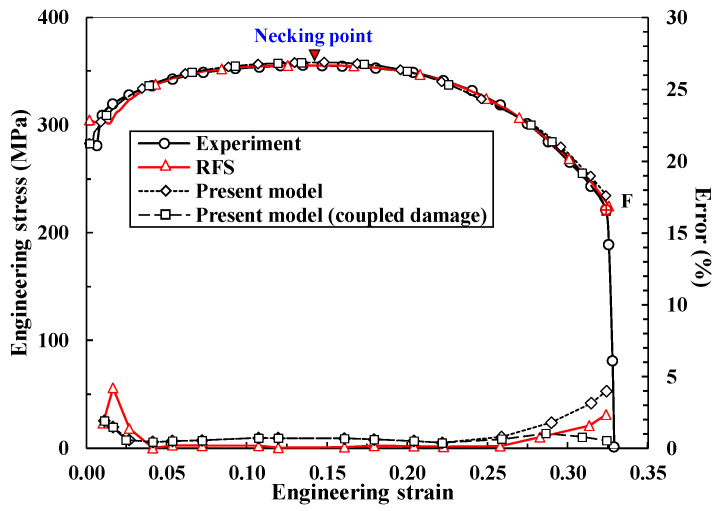
Comparison of the predictions of our damage-coupled model, the RFS, and the coupling-free model with the experimental tensile test.

**Figure 9 materials-14-04876-f009:**
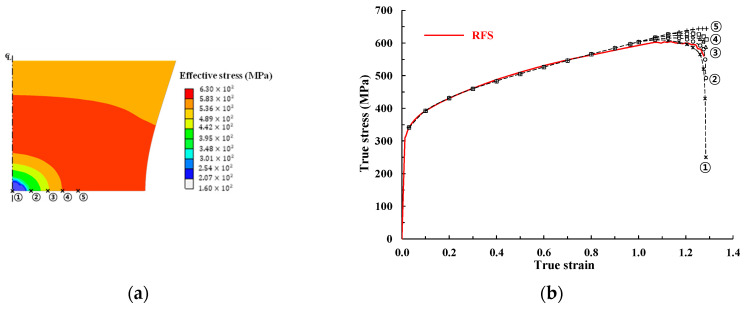
Effect of damage and initiation of fracture. (**a**) Definition of selected points and the effective stress at the fracture instant; (**b**) change of effective stresses at the selected points with true strain.

**Figure 10 materials-14-04876-f010:**
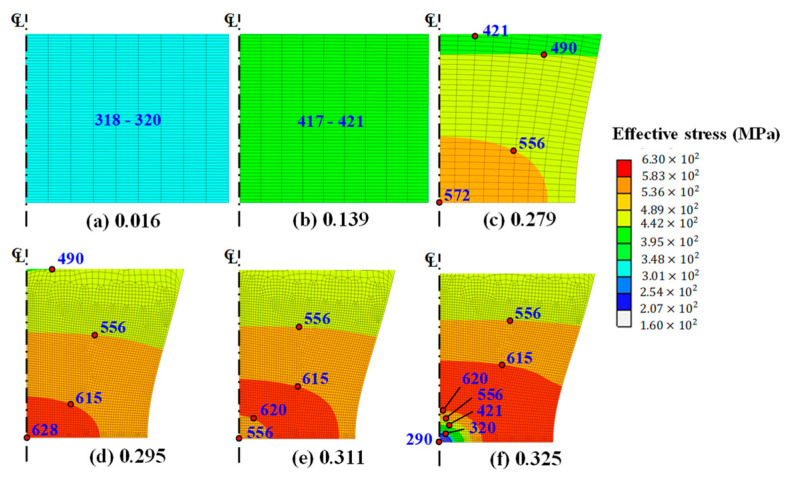
Change of effective stress (MPa) with engineering strain in the necking region with an emphasis on damage effect.

**Figure 11 materials-14-04876-f011:**
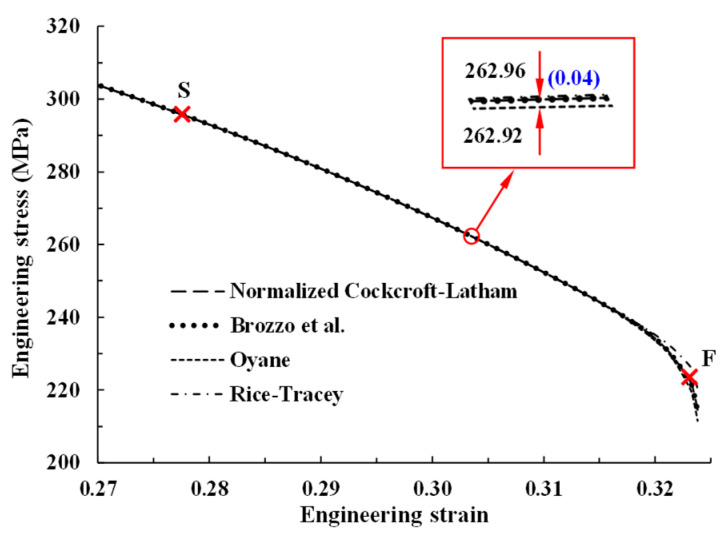
Tensile behavior predictions based on the softening and fracture points S and F, respectively, for the present model with different damage models [52,53,54,55].

**Figure 12 materials-14-04876-f012:**
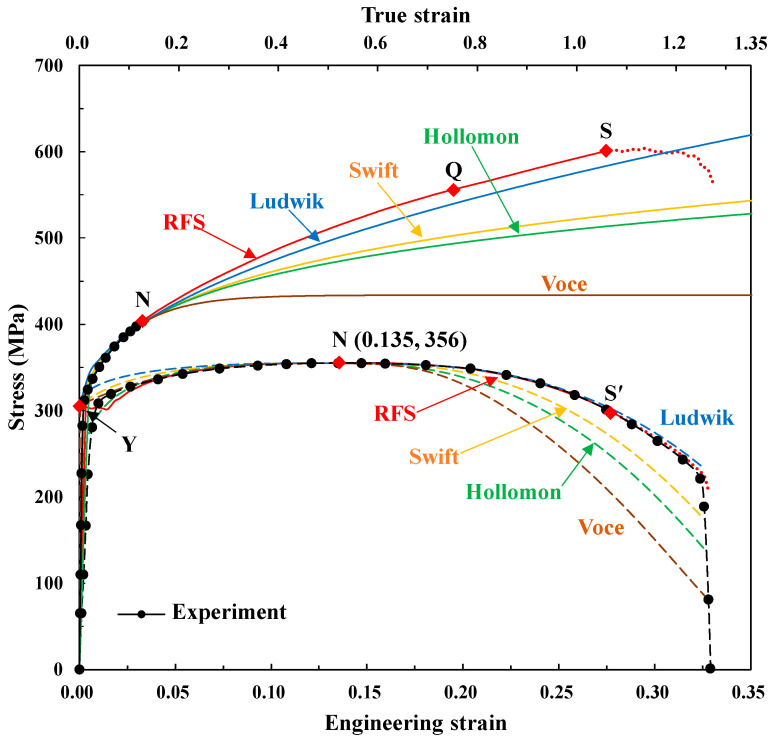
Comparison of flow stresses and their corresponding tensile test predictions of the four fundamental flow models with an emphasis on necking point and yield strength.

## Data Availability

Not applicable.

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
