# Peer review of "A Novel Flow Model of Strain Hardening and Softening for Use in Tensile Testing of a Cylindrical Specimen at Room Temperature"

_materials, 2021, doi:10.3390/ma14174876_

Round 1
Reviewer 1 Report
This manuscript presented a new flow stress model based on the Swift method, which is rather simple and can taking account strain hardening and damage softening, hence could be used in the cases involving large plastic deformation. It was well written, and the conclusions were convincing. Some minor revision is suggested.
- In the beginning of Paragraph 3 of “Introduction”, “Spite of” should be revised by “In spite of”.
- In Line 5 of Section 2, “lower” seems should be revised as “upper”.
- In the paragraph above Fig. 8, the authors wrote “p = 0.85 was calculated based on engineering strains of 0.277 and 0.327 at S and F”, but it was not clear how was the calculation implemented?
- The meaning of Fig. 9 needs further explanation: what was the meaning of the colors? Why different position possesses different true stress – true strain curve?
- The meaning of the caption of Fig. 10 is hard to be understood.
6. It is suggested to compare the FE simulated strain distribution in uniaxial tension with the experimental results to further verify the effectiveness of the proposed flow stress model.
Author Response
We would like to thank the reviewer for the valuable comments and very helpful suggestions. We carefully revised the manuscript according to the reviewer's advice. Also, we addressed the Reviewer's comments point by point. The revision made in the manuscript is highlighted in yellow.

Reviewer 2 Report
the results presentation should be improved to lead a better understanding, espacially for the new model presentation, from introduction to conclusion, one don't know what is your new model, what are the input-output parameters of your model, what is the validity of your model and it applicaiton case, in title you have indicated one new model, in conclusion part, you mensioned two model :
- in introduction part, it is necessary to indicate explicitely the originality of the study, the main acheivement of the new model, the application domain and its validity ...
- in material and methods part, it is necessary to give a minimum information about the experimence (alloy composiiton, sample dimension, test condition etc...); for method presentation, it is necessary to indicate clealy what is the new point, what are the two studied "cases", discuss espacially the validity of new merhod;
- in caption of figure 1, it is necessary to precise the used alloy; in the presetation of the figure 1, it is more easy to use the point name N, O S and F instead of digital number;
- in caption of figure 2, 3 and 6, it is necessary to precise what are the case 1 and case 2;
- in cpation of figure 4 and 5, it is necessary to precise the physical meaning of all the symbols;
- in caption of figure 7, it is necessary to precise what is alpha 2;
- in caption of figure 8, it is necessary to precise the used new model presented;
- in caption of figure 9 and 10, it is important to indicate how all results have obtained and with which model;
- in caption of figure 11, it is important to precise what curve has been obtained with your new model.
- in conclusion part, it is important to precise explcitely what are the two new models.
Author Response
We would like to thank the reviewer for the valuable comments and very helpful suggestions. We carefully revised the manuscript according to the reviewer's advice. Also, we addressed the Reviewer's comments point by point. The revision made in the manuscript is highlighted in yellow.
Please see the attachment.

Reviewer 3 Report
- In equations (1) and (2), explain what stresses and deformations are used.
- It is recommended to add a more detailed description of obtaining of experimental data in Figure 1.
- It is recommended to add a more detailed description of finite element modeling in Figures 9 and 10. What problem was solved? What is the type of finite elements? How was the proposed mechanical model of the material behavior implemented? What are the boundary conditions? You need to add scale bars. What do the different curves in Figure 9 show?
- It is recommended to add a discussion of the applicability of the proposed model. For what engineering tasks will the use of this model give a significant result? How practical is it to use this model? How easy is it to experimentally obtain model parameters for different types of materials? And how easy is it to use this model in numerical modeling of engineering problems?
Author Response

(The authors gave the same response as above.)

Reviewer 4 Report
The paper deals with flow curve modeling using metallic round specimens for bulk forming. The core of the paper is the extension of the well-known Swift model with the aim to be able to represent the post-necking behavior. Firstly, a second term is added which can adjust the slope of the yield curve model for larger strains. Secondly, one parameter is extended so that softening due to damage can be represented.
The approach to modeling material behavior over the full range of tensile testing seems interesting. Nevertheless, some questions arise on which the authors should comment:
- Figure 1 shows a yield curve based on an experiment (RFS). Why is there a need to interpolate these data points using a model? Why is it not possible to use the data points, possibly with local non-linear interpolation, for a numerical analysis? In this way, the modeling error caused by the parameter determination of the model can be avoided.
- If, apart from the previous point, modeling is desired or necessary, the question arises to what extent the presented model has the necessary flexibility to represent flow curves of different materials. For the standard modeling of flow curves, various approaches exist besides Swift, e.g. Voce, Gosh, Hockett-Sherby. Depending on the material, these approaches represent the material response differently well. Accordingly, it is to be expected that the modification of a single model does not have the flexibility to represent different material classes. This significantly reduces the area of application of the model shown. Consequently, a corresponding extension of further models would be interesting.
- Overall, the paper lacks motivation as to where the advantages lie over existing models.
- The paper is based on the publication by Joun et al [10]. Unfortunately, I did not have access to this publication. Without this basis it is partly difficult to follow the paper.
- The paper does not mention anything about the test and measurement method, which makes it impossible to assess the experimentally determined values.
- The standard conversion between engineering (εe) and true strain (εt) is usually εt = ln(1+ εe). However, a different type of relationship is found between the two curves in Figure 1, which cannot be understood without further explanation.
- The Swift parameter for the flow curve up to the necking is determined by means of 2 points (P, N). As a result it follows that this approach strongly depends on the choice of the first point (P). Why are not all available points used for an interpolation of this area?
- There is a lack of basic information about the FEM used to investigate the models.
- When investigating failure, model reduction by modeling a quarter model should be avoided, since the failure behavior is generally not symmetrical.
- Minor and formal comments:
- The caption for figure 2 should be below the figure.
- In the caption of figure 4 the number 2 should be placed as index
Author Response
We would like to thank the reviewer for the valuable comments and very helpful suggestions. We carefully revised the manuscript according to the reviewer's advice. Also, we addressed the Reviewer's comments point by point. The revision made in the manuscript is highlighted in yellow. Please kindly refer to the attached file.

Round 2
Reviewer 1 Report
The manuscript has been properly revised, so it is recommended for publication.
Author Response
Thank you very much for the valuable comments and very helpful suggestions. We are very happy to have received a positive evaluation from the reviewer.
Reviewer 2 Report
any comments and remarks from reviewers have been considered in the 2nd version of the manuscript.
First of all, in introduction part, reader can not obtain easily information about the orignality of this work and about the main achievement.
Secondly, all captions of figures should be understanding without looking for all details between lines and between words;
Thirdly, there are always important confusion bewteen the title (a novel model) and the conclusion (Two representative models).
Author Response

(The authors gave the same response as above.)

Reviewer 3 Report
The manuscript can be published in this form.
Author Response

(The authors gave the same response as above.)

Round 3
Reviewer 2 Report
in the revised version of the manuscript, author has made necessary corrections includding most of remarks from reviewers.
Just a minor remark to lead a better understanding: in the manuscript, when you talked about all different critical points (N, S, F, case 1 and 2 in swift model) it is important to cite your figure 1 espacially each time when it is possible to guide the readers.
Author Response
Thank you very much for the valuable comments and very helpful suggestions. We are very happy to have received a positive evaluation. We really appreciate the time and effort that the reviewer has dedicated to providing valuable feedback. We have highlighted the changes within the manuscript.